# Hepatocellular Carcinoma and Hepatitis: Advanced Diagnosis and Management with a Focus on the Prevention of Hepatitis B-Related Hepatocellular Carcinoma

**DOI:** 10.3390/diagnostics13203212

**Published:** 2023-10-14

**Authors:** Soo Ryang Kim, Soo Ki Kim

**Affiliations:** Department of Gastroenterology, Kobe Asahi Hospital, Kobe 653-0801, Japan; asahi-hp@arion.ocn.ne.jp

**Keywords:** hepatitis B virus, hepatocellular carcinoma, nucleos(t)ide analogues, secondary prevention, HBV DNA, immune-tolerant phase, hepatitis B virus surface antigen, hepatitis B virus core-related antigen, covalently closed circular DNA

## Abstract

Though the world-wide hepatitis B virus (HBV) vaccination program has been well completed for almost thirty years in many nations, almost HBV-related hepatocellular carcinoma (HCC) occurs in unvaccinated middle-aged and elderly adults. Apparently, treating 80% of qualified subjects could decrease HBV-related mortality by 65% in a short period. Nevertheless, globally, only 2.2% of CHB patients undergo antiviral therapy. The HBV markers related to HCC occurrence and prevention are as follows: the HCC risk is the highest at a baseline of HBV DNA of 6–7 log copies/mL, and it is the lowest at a baseline of an HBV DNA level of >8 log copies/mL and ≤4 log copies/mL (parabolic, and not linear pattern). The titer of an HBV core-related antigen (HBcrAg) reflecting the amount of HBV covalently closed circular DNA (ccc DNA) in the liver is related to HCC occurrence. The seroclearance of HBs antigen (HBsAg) is more crucial than HBV DNA negativity for the prevention of HCC. In terms of the secondary prevention of hepatitis B-related HCC involving antiviral therapies with nucleos(t)ide analogues (NAs), unsolved issues include the definition of the immune-tolerant phase; the optimal time for starting antiviral therapies with NAs; the limits of increased aminotransferase (ALT) levels as criteria for therapy in CHB patients; the normalization of ALT levels with NAs and the relation to the risk of HCC; and the relation between serum HBV levels and the risk of HCC. Moreover, the first-line therapy with NAs including entecavir (ETV), tenofovir disoproxil fumarate (TDF), and tenofovir alafenamide (TAF) remains to be clarified. Discussed here, therefore, are the recent findings of HBV markers related to HCC occurrence and prevention, unsolved issues, and the current secondary antiviral therapy for the prevention of HBV-related HCC.

## 1. Introduction

Chronic hepatitis B (CHB) is the most common reason for the development of hepatocellular carcinoma (HCC) and the second main reason of carcinoma-associated death worldwide [1,2,3]. Approximately 3.9% of the world’s population, representing 292 million people, has been chronically infected with hepatitis B virus (HBV) (CHB) since 2016, with CHB being responsible for almost one million mortalities every year [4,5,6].

Global mortalities from HCC related to HBV are estimated to increase by two times by 2040 [2,3,7]. A careful examination of randomized or matched-control studies shows that long-term treatment with nucleos(t)ide analogues (NAs), including entecavir (ETV, Bristol Myers Squibb, New York, NY, USA), tenofovir disoproxil fumarate (TDF, GlaxoSmithKline, London, UK), and tenofovir alafenamide (TAF, Gilead Sciences, Foster, USA), reduces the incidence of HCC [8,9,10].

Though the universal HBV vaccination project has been well performed for about thirty years in many nations, most HBV-associated HCCs develop in non-vaccinated middle-aged and older adults [11,12]. Therefore, considering the birth cohort result on HBV-related HCC frequency, the secondary prohibition of HCC with an antiviral treatment for subjects who are already chronically infected with HBV is the main method of decreasing HBV-associated mortality. It has been suggested that treating 80% of qualified patients could decrease HBV-related deaths by 65% in a short period [5]. Nevertheless, less than 10% of qualified subjects with CHB experienced antiviral treatment in 2015 [7]. Regarding the secondary prohibition of hepatitis B-related HCC with antiviral therapies, unsolved issues remain in terms of the definition of the immune-tolerant phase; the optimal time for starting antiviral therapy with NAs within the limits of high ALT levels as criteria for the therapy of CHB patients; the normalization of ALT levels with NAs and the relation to the risk of HCC; the relation between serum HBV markers, such as HBV DNA, HBs antigen (HBsAg), and HB core-related antigen (HBcrAg), and the risk of HCC; and the effectiveness of initial therapy with NAs, including ETV, TDF, and TAF for chronic hepatitis B to prohibit HCC. Therefore, discussed here are the issues stated above; the current antiviral therapies for the prevention of HBV-related HCC are also reviewed.

## 2. Definition of the Immune-Tolerant Phase

The immune-tolerant stage showing the early stage of CHB is of great interest but not well known. The idea of true immune tolerance has not been sufficiently studied from the immunological point of view. Chief international guidelines such as the American Association for the Study of Liver Diseases (AASLD), the European Association for the Study of the Liver (EASL), and the Asian Pacific Association for the Study of the Liver (APASL) have not yet attained an agreement on the definition of the immune-tolerant stage. The recent guidelines issued in 2016 by the AASLD define the immune-tolerant phase in a more traditional manner, i.e., using normal alanine aminotransferase (ALT; <30 U/L in males and <19 U/L in females as upper limits of normal (ULN)), than through a local examination of criteria ranges, a high HBV DNA (typically above 1 million IU/mL), a positive hepatitis B e-antigen (HBeAg), and minimum inflammation and fibrosis demonstrated in liver histopathology [13,14]. This is in contrast to another HBeAg-positive case, the immune-active stage, with the point distinguishing characteristics of a high serum ALT, but probably not as high as the serum HBV DNA (>200,000 IU/mL), and extreme to severe inflammation or fibrosis demonstrated in liver histopathology [14].

On the other hand, the practice guidelines of the APASL, updated in 2015, define the immune-tolerant stage slightly differently from the AASLD, chiefly using a different threshold of serum HBV DNA (>20,000 IU/mL) and an intruding age as some of the criteria (typically below 30 years) [15].

The idea of true immune tolerance has been questioned because immunological findings have shown that children and young subjects with CHB maintain an immune feature that is less compromised than that seen in older patients [16].

The EASL practice guidelines issued in April 2017 put this stage of a new nomenclature—Stage 1 or HBeAg-positive chronic HBV infection—in place of the routine immune-tolerant stage [8]. The features of this stage include what the AASLD guidelines write, together with a few special characteristics at the molecular and immunological levels, as high-level HBV DNA integration and clonal hepatocyte expansion [17], possibly preceding hepatocarcinogenesis while keeping HBV-specific T cell function at least until young adulthood, with a very low rate of natural HBeAg loss but remaining highly infectious due to the high levels of HBV DNA [8]. Together, a positive hepatitis B e-antigen (HBeAg), a high serum HBV DNA, and normal serum alanine aminotransferase (ALT) levels are the three important characteristics of the immune-tolerant stage across the three major international guidelines. No consensus, however, has been obtained about the lower cut-off level of HBV DNA, which changes between 6 log10 IU/mL and 2 × 7 log10 IU/mL for defining this phase in clinical practice guidelines [1,8,15,18,19]. Although recent guidelines recommend against initiating antiviral therapy for immune-tolerant CHB subjects, some recent data indicate that treating such subjects might decrease the progression of liver fibrosis and the risk of hepatocellular carcinoma [13].

The remaining references on the HCC risk in immune-tolerant CHB are few. A Korean cohort study of 413 HBeAg-positive CHB subjects with normal ALT levels (AASLD criterion), high HBV DNA levels (≥20,000 IU/mL), and no data on liver cirrhosis was compared with another cohort of 1497 CHB patients in the immune-clearance stage treated with NA [20]. During the long-term follow-up, the 10-year predicted cumulative incidence of HCC was significantly higher in the untreated immune-tolerant subjects than in the treated immune-clearance phase subjects (12.7% vs. 6.1%; *p* = 0.001) [20]. These data, with different conclusions from the above data, may indicate that either of these firstly immune-tolerant patients have evolved over time, or that NA therapy decreased the risk of HCC to the extent that it would be even lower than in the untreated immune-tolerant patients [13]. The data from another nationwide real-life Korean study of 484 HBeAg-positive CHB patients with normal ALT levels (<40 U/L), high HBV DNA levels (>20,000 IU/mL), and no cirrhosis throw some light on this important yet difficult-to-analyze topic. A total of 87 of the 484 subjects received NA compared with 397 untreated subjects in the control group; NA therapy significantly decreased the risk of HCC (with an adjusted odds ratio of 0.189; *p* = 0.004) [21].

## 3. When to Start Antiviral Therapies with NAs in Terms of ALT

Almost all of the recent clinical guidelines for CHB have the same recommendations for antiviral therapy [8,15,18,22,23] based on high serum values of alanine aminotransferase (ALT) and HBV DNA in CHB patients with no liver cirrhosis. Though the European clinical practice guidelines recommend that subjects with HBeAg-positive CHB may be treated if they are older than 30 years and have high HBV DNA values irrespective of constantly normal ALT levels, the evidence level is significantly low (evidence level III; grade 2 of recommendation) [8]. Almost all other clinical guidelines recommend that antiviral therapy be postponed until the patients demonstrate significantly high values in ALT levels or evidence of inflammation or fibrosis in histopathology [8,15,18,19]. When the goal of antiviral therapy is more the prohibition of HCC than the management of hepatic inflammation or fibrosis, the guidelines should be considered with carefulness while thinking that HBV-related hepatocarcinogenesis could be underway without the signs of significant hepatic inflammation and/or fibrosis [20,24,25,26,27,28]. Thus, the recently recommended timeline for antiviral therapy based on ALT values may be inapplicable for the efficacious prevention of HCC [11,12,20,28].

In context with the data from these preclinical investigations and the clinical data, an extreme serum HBV DNA level (5–7 log10 IU/mL) was thought to be a risk reason for significant hepatitis in CHB patients irrespective of normal ALT values and the absence of significant fibrosis [29]. In addition, HBV DNA integration into the patients’ chromosomes could be ongoing in HBeAg-positive patients with a long-lasting chronic HBV infection, which may more additionally upgrade chromosomal instability followed by the functional loss of tumor-suppressor genes or the activation of tumor-promoting genes engaged in hepatocarcinogenesis [4,17,30]. These results suggest that the currently recommended timeline for antiviral therapy based on ALT elevation is not optimal for preventing HCC [1,12,20,28].

## 4. Limitations of Elevated ALT Levels as Criteria for Therapy of CHB Patients

The serum ALT value is commonly adopted as a marker of active necroinflammation in the liver because of its convenient measurability, given that liver histopathology is not capable of assessing the severity of hepatitis in most patients. Recent treatment guidelines assume a prevailing hypothesis that a normal ALT level is indicative of an inactive liver disease [8,15,18,19]. On the contrary, antiviral therapy may be postponed until subjects show an increase in ALT levels [26]. Indeed, non-cirrhotic patients with normal ALT levels are generally advised not to start antiviral therapy irrespective of relatively high values of HBV DNA, unless they are thought to have significant liver disease assessed through liver histopathology or unless they have a family history of cirrhosis or HCC. However, this method has lately been doubted.

Firstly, the serum ALT levels are nonsensitive markers of HBV-associated hepatocyte injury. Former data demonstrated that a significant size of subjects with continuously normal ALT levels display severe hepatic necroinflammation and/or fibrosis [31,32,33,34,35]. Furthermore, group studies have also shown that patients with high HBV DNA levels may advance to HCC or end-stage liver disease even without a significant increase in ALT levels, irrespective of HBeAg [2,20,28].

Secondly, the unclearness of normal ALT levels is a problem that needs to be investigated more. Generally, guidelines often adopt the term ULN when indicating ALT levels. This ULN value for ALT has historically been considered 40 IU/L, irrespective of gender [36]. Perfectly, the normal extent of ALT levels should be decided in person without clear liver disease. As mentioned previously, some subjects with an ALT level of ≤40 IU/L, which has been commonly admitted as the threshold of a normal range, were thought to have active liver disease with significant inflammation and fibrosis.

The size of patients with significant histopathology was actually found to be smaller when using the lower ULN of the ALT: 30 IU/mL for males and 19 IU/mL for females [37]. Another study from Hong Kong demonstrated that subjects with ALT values of <0.5 × ULN have a significantly lower risk of cirrhotic morbidity and HCC compared with patients who have ALT levels between 0.5 and 1 × ULN [38,39]. Accordingly, instead of the conventional ULN of ALT (40 IU/L), several normal alternative levels have been indicated. A Korean group proposed normal ALT levels of 33 IU/L for males and 25 IU/L for females based on 1105 biopsy-proven normal livers [36]. The former AASLD guidelines take a stricter ULN value of ALT (30 IU/L for males and 19 IU/L for females) [14]; on the other hand, the latest AASLD guidelines revert to the earlier ULN value of ALT (35 IU/L for males and 25 IU/L for females) [18]. However, the EASL and APASL guidelines still adopt 40 IU/L as the ULN value of ALT, or advocate the use of local laboratory criteria [8,15]. Taken together, lowering the ULN of ALT levels should be thoughtfully considered in therapy decisions to prevent additional liver-associated complications such as HCC.

## 5. Relation between Normalization of ALT with NAs and Risk of HCC

The literature on the relation between the efficacy of NAs for preventing HCC in CHB patients and the normalization of ALT levels is also scarce. One Korean study on antiviral therapy aimed at preventing HCC in CHB patients associated with normalized ALT levels demonstrated that 610 patients with chronic hepatitis B received ETV or TDF between 2007 and 2017. The subjects were classified into an ALT-normalized group (group 1) and a non-normalized group (group 2) within a year of potent antiviral therapy. The death ratio and HCC frequency were investigated in every group. The number of subjects who showed ALT normalization at 1 year of therapy was 397 (65.1%) of 610. During a median follow-up duration of 86 months, 65 of the 610 (10.7%) patients developed HCC. The total HCC frequencies in group 1 and group 2 were 4.8% and 13.6% at 5 years (*p* = 0.001) and 6.8% and 16.9% at 8 years, respectively (*p* < 0.001). The total occurrence of HCC was significantly lower in group 1 than in group 2 (*p* < 0.001), suggesting that a normalized ALT within 1 year of starting antiviral medicines decreases the risk of developing HCC [40].

Another Korean study on 4639 subjects diagnosed with CHB started therapy with ETV or TDF. Landmark and time-dependent Cox analyses revealed a normal ALT at <35 U/L in males and <25 U/L in females, and they revealed the virological response (VR) as serum hepatitis B virus DNA as <15 IU/mL. During a median follow-up period of 5.6 years, 509 of the 4639 (11.0%) subjects had progressed HCC. The ALT levels were normalized in 65.6% at 1 year and in 81.9% at 2 years and were related to a significantly lower HCC risk as determined through landmark (*p* < 0.001) and time-dependent Cox analyses (adjusted odds ratio 0.57; *p* < 0.001). Compared to ALT normalization within 6 months, postponed ALT normalizations at 6–12, 12–24, and >24 months were related to a significantly increased HCC risk (AHR 1.40, 1.74, and 2.45, respectively; *p* < 0.001), irrespective of steatosis or cirrhosis at the baseline and VR during therapy.

Contrastingly, neither earlier VR (AHR 0.93; *p* = 0.53) nor earlier hepatitis B e-antigen seroclearance (AHR 0.91; *p* = 0.31) were related to a significantly lower HCC risk [41]

## 6. Relation of Serum HBV DNA Levels and HBV DNA Genotypes and Subgenotypes with the Risk of HCC

The relation between the baseline HBV DNA values and the risk of HCC has been regarded as positively linear under a natural group study on untreated CHB subjects (REVEAL-HBV study), demonstrating that the risk of HCC is the highest in subjects with baseline HBV DNA values of >106 copies/mL (~5 log10 IU/mL) [24]. The investigation has limits, however, in extrapolating to HBeAg-positive CHB subjects with high viral loads, because most of the subjects in the REVEAL group were HBeAg-negative (85%), and HBV DNA titers >106 copies/mL were not separately examined. Accordingly, the real risk of HCC in patients with higher levels of HBV DNA (>106 copies/mL) remains unclarified. Indeed, following the data from the same group demonstrated that subjects with HBV DNA values of >107 copies/mL had a significantly lower risk of HCC than those with HBV DNA values between 106 and 107 copies/mL [25,41,42]. Accordingly, in a recent historical group study including 6949 non-cirrhotic patients without significant ALT increases (<2 × ULN at least for 1 year) irrespective of HBeAg positivity, the relation between the HBV DNA levels and HCC risk was not linear, but parabolic, demonstrating the highest HCC risk with extreme serum HBV DNA values of 5.0–7.0 log10 IU/mL and demonstrating the lowest HCC risk with HBV DNA levels of >8 log10 IU/mL and ≤4 log10 IU/mL, irrespective of the ALT levels [2]. A currently issued multicenter historical group investigation has also evinced a low risk of HCC occurrence in untreated non-cirrhotic subjects with an HBV DNA level of > 107 IU/mL [12,43].

Moreover, patients in the “grey-zone”, defined as those with no cirrhosis, persistently normal ALT levels, and moderate levels of serum HBV DNA (between 4 log10 IU/mL and 8 log10 IU/mL), have been considered to be at a significantly higher risk of developing HCC if left untreated compared with patients in the gray zone treated with anti-HBV drugs [2,20]. Notably, a further large-scale multicenter cohort study supported those findings [1,4].

Compared to patients showing high baseline HBV DNA values of ≥8.00 log10 IU/mL, those showing values of 7.00–7.99, 6.00–6.99, and 5.00–5.99 log10 IU/mL had 2.48, 3.69, and 6.10 times higher adjusted risks of HCC, respectively, while under persisting therapy. The inverse relation between the baseline HBV DNA levels and on-therapy HCC risk was constantly shown in unadjusted, multivariable-adjusted, propensity score (PS)-weighted, PS-matched, sensitivity, and competing risk examinations in the whole group and in the several subgroups of subjects. In addition, the HCC risk of subjects with an extreme viral load and under primary antiviral therapy was significantly lower than that of untreated subjects with the same extent of HBV DNA levels; however, it was significantly higher than that of patients showing high viral loads, showing that antiviral therapy could decrease the risk of HCC in extreme viral load groups, but could not revert to the levels of high viral load groups [4]. Accordingly, the traditional concept of “linear association” between HBV DNA and the risk of HCC may have to be changed to “parabolic association” [42].

The relation between the baseline HBV DNA values and ongoing-therapy HCC risk has remained unclarified. Recent findings are in line with former studies on untreated HBeAg-positive subjects, showing that lower baseline HBV DNA values (but above 5 log10 IU/mL) are related to a significantly higher risk of HCC during the follow-up without therapy [2,20,44]. Nevertheless, the current multicenter group investigation offers a new observation that the baseline HBV DNA levels have a significant relation to the risk of HCC even during long-duration therapy with potent antiviral medicines. The inverse relation between the baseline HBV DNA levels and the risk of HCC persists for up to 10 years of persistent potent antiviral therapy in HBeAg-positive subjects with CHB [4].

The serum HBV DNA values change with the interaction between the human and virus during the natural course of CHB. Almost all of the subjects with CHB have very high values (≥8 log10 IU/mL) of HBV DNA at the start phase of the infection [26]. The immune-related destruction of HBV-infected hepatocytes via HBV-specific T-cells may, however, result in clonal occurrence and the expansion of HBV-resistant hepatocytes that can escape such immune defense mechanisms [45,46,47]. Thus, a decrease in the HBV values (e.g., <8 log10 IU/mL) might show the accumulation of hepatocyte injury, changes in the hepatocyte tissues, and a following upgrade in the risk of HCC [17,30,46,48,49]. In addition, HBV DNA integration into human chromosomes could be ongoing even in HBeAg-positive subjects with early-stage chronic HBV infection regarded as immune-tolerant and may additionally upgrade chromosomal instability [17]. The random integration of the viral genome into the human chromosome may lead to the functional loss of tumor suppressor genes or the activation of tumor-promoting genes specifically engaged in hepatocarcinogenesis [30,50]. These findings offer a necessity for early therapy intervention based on the HBV DNA value in subjects with CHB before the occurrence of irreversible changes including clonal hepatocyte expansion and HBV DNA integration into human chromosomes. Thus, non-cirrhotic grown-up subjects with extreme values of HBV DNA (4.0–8.0 log10 IU/mL) should be considered for antiviral therapy irrespective of their ALT values to additionally decrease the occurrence of HCC [12].

The influence of HBV genotypes and subgenotypes on HCC has been reported in eight genotypes of HBV (A through H), which differ from each other in terms of viral genome sequence by more than 8%, and multiple subgenotypes, which differ from each other by 4–8%. Lately, studies investigating the relation between the risks of proceeding HCC and cirrhosis via special HBV genotypes and subgenotypes have reported marked differences in result. Certain HBV genotypes and subgenotypes, such as genotypes C, B2-5, and F1, seem to be related to a higher risk of occurring HCC, and others, such as genotypes B1, B6, and A2, appear to be related to a lower risk of morbidity of HBV. Their understanding of the roles of HBV genotypes and subgenotypes in the result of HBV infection is limited, as few inhabitants-based prospective investigations have been completed, and most investigations only compare the result in regions where two genotypes are overwhelmed, whereas others have not studied subgenotypes [51].

Recently, Japanese researchers reported that the frequency of occurrence of hepatocellular carcinoma in Japanese patients infected with hepatitis B virus is the same between genotypes B and C in the long-term. Previously, HBV/C infection was related to more severe disease progression, showing as proceeding to cirrhosis and HCC, than HBV/B infection. However, thereafter, HCC related to HBV/B increased, and no significant difference was evinced between HBV/B and HBV/C. HCC occurrence was consistently demonstrated even in HBV/B infection, particularly among old subjects with severe fibrosis compared to HBV/C. HBV/B-infected subjects developed HCC later in life, and in the long-term, they showed no differences in the frequency of HCC occurrence in the ratio between these two genotypes [52].

## 7. Relation between HBV Virus Markers: HBsAg, HBcrAg, and HCC

The viral covalently closed circular DNA (cccDNA) is related to the continuity of the infection in hepatocytes. To successfully control the subject therapy and follow-up, and to advance new antiviral therapy directly targeting the intrahepatic pool of cccDNA, serum substitute markers including viral activity in the liver are imperatively needed. It has been demonstrated that the quantification of HBcrAg in serum relates to cccDNA quantity and activity and could be adopted to observe disease progression [53].

One of the causes for HCC occurrence with NA therapy is the difficulty of these drugs in eradicating HBV-cccDNA in the liver [54,55]. In particular, these NAs just inhibit the reverse transcription of HBV-RNA into HBV-DNA but do not directly inhibit the transcription and subsequent protein synthesis from HBV-cccDNA [56]. Since the amount and transcriptional activity of intrahepatic HBV-cccDNA is regarded as influencing the frequency of HBV-related HCC with NA therapy [57], the quantification of intrahepatic HBV-cccDNA should be a critical sign for prohibiting HCC. However, the quantification of HBV-cccDNA needs an invasive liver biopsy. [54].

It has become slowly clear that NA treatment does not thoroughly remove and prohibit the development of HCC [58]. The serum HBV-DNA value, the most essential biomarker for evaluating the risk of HCC occurrence [24,38], is no longer as important as NAs in suppressing HBV-DNA; therefore, accurate biomarkers other than HBV-DNA that predict HCC development are urgently needed [54].

In the subgroup of HBeAg-negative subjects with HBV DNA values between 2000 and 19,999 IU/mL (intermediate viral load (IVL)) and normal values of ALT, HBcrAg values ≥ 10 KU/mL have indicated patients to be at an increased risk of HCC (odds ratio, 6.29; confidence interval, 2.27–17.48). The risk of HCC in patients with an IVL and a high value of HBcrAg does not differ significantly from that in subjects with a high viral value (≥20,000 IU/mL) [59].

Subjects with an IVL but a low value of HBcrAg demonstrate a low risk of HCC, with a yearly occurrence of 0.10% (95% confidence duration, 0.04–0.24%). In a long-duration follow-up study of 2666 subjects with chronic HBV infection (genotypes B or C), the value of HBcrAg signals an independent risk factor of HCC. In addition, an HBcrAg value of 10 KU/mL indicates subjects with an IVL and at a high risk for HCC [59].

Recently, non-invasive surrogate markers of quantitative HBsAg and HBcrAg assays were developed [60,61], and an analytical implementation of an immunoassay for total antigens including complexes via pretreatment with (iTACT)-HBcrAg was confirmed. The sensitiveness of iTACT-HBcrAg (2.1 log U/mL) was about 10 times greater than that of a conventional HBcrAg assay (2.8 log U/mL). (i) With the use of iTACT-HBcrAg, HBcrAg was seen in the sera of 97.5% (157/161) of subjects with CHB, of whom 75.2% (121/161) had ≥2.8 log U/mL HBcrAg and 22.4% (36/161) had 2.1–2.8 log U/mL HBcrAg that was unmeasurable using G-HBcrAg. (ii) Also, with the use of iTACT-HBcrAg, 9 and 2 of 13 HBV-reactivated subjects were HBcrAg-positive before and had HBV DNA positivity, respectively, and 7 and 4 patients were HBcrAg-positive before and had HBsAg-positivity via an immune complex transfer chemiluminescent enzyme immunoassay, respectively. (iii) The HBcrAg examined via iTACT-HBcrAg before HBV reactivation was kept in empty particles (22 KDa precore protein) [62].

iTACT-HBcrAg was adopted to better monitor the response to anti-HBV therapy in HBeAg-negative subjects and for the early discovery of HBV reactivation [62].

The serum HBcrAg and HBsAg values are substitute signs of intrahepatic covalently closed circular DNA. The measuring extent of the recent HBcrAg assay is comparatively confined. The capability of HBcrAg and HBsAg examined via ultrasensitive assays for estimating HCC occurrence in subjects with CHB treated with ETV was assessed. A retrospective group investigation of 180 subjects who were given ETV for >1 year was conducted. All subjects showed a negative hepatitis B e-antigen at the start line. The serum HBcrAg and HBsAg values at the start line and at year 1 were examined in every subject via ultrasensitive assays adopting iTACT technology. During the medium follow-up of 11.0 years, 22 patients developed HCC (11.8/1000 person years). The start-line HBsAg values were not related to HCC development during the ETV treatment. On the other hand, high HBcrAg values at the start line and at year 1 were significantly related to HCC occurrence (log-rank test; *p* < 0.001). In 110 patients (61.1%) with ≥4.0 log U/mL at the start line (high HBcrAg group), the HBcrAg value deceased to ≤2.9 log U/mL at year 1 in 25 patients (22.7%). The adjusted odds ratio for HCC occurrence was significantly lower in patients with HBcrAg ≤ 2.9 log U/mL at year 1 than in those in the high HBcrAg group. Conclusively, an examination of the HBcrAg value via an ultrasensitive measure was considered to be more capable for estimating HCC during antiviral therapy than the recent HBcrAg assay [63].

Seventeen (5 HCC and 12 non-HCC) subjects with CHB who attained HBsAg seroclearance, defined by the former assay with the use of an Architect HBsAg QT kit, were enrolled. The HBsAg and HBcrAg values were assessed in their stored serum samples by using ultra-highly sensitive assays characterizing iTACT technology.

The five HCC patients were positive for HBsAg or HBcrAg via iTACT-HBsAg or iTACT-HBcrAg at all follow-up points. The HBcrAg values in the HCC group, assessed via iTACT-HBcrAg, were significantly higher than in the non-HCC group at HBsAg seroclearance (3.6 logU/mL (2.8–4.2) versus 2.6 (<2.1–3.8), *p* = 0.020). The best cutoff value of iTACT-HBcrAg for estimating HCC occurrence was 2.7 logU/mL by the operator running a feature curve analysis. The occurrence of HBcrAg ≥ 2.7 in the HCC group was significantly higher than that in the non-HCC group (100% (5/5) versus 33% (4/12); *p* = 0.029) [64].

The above data suggest that a remaining low level of viral antigen predicts HCC occurrence even when HBsAg seroclearance is attained with the use of conventional assays. The data also show that the iTACT assays for HBsAg and HBcrAg would be important in managing CHB patients [64].

Recently, one promising paper reported the inhibition of hepatitis B virus via AAV8-derived CRISPR/SaCas9 expressed from liver-specific promoters to clear cccDNA [65]. The authors analyzed the liver particularity of several promoters and constructed candidate promoters in the CRISPR/Staphylococcus aureus Cas9 (SaCas9) system combined with hepatotropic AAV8 (whereby AAV refers to adeno-associated virus) to verify the effectiveness against HBV. The data demonstrated that the reconstructed CRISPR/SaCas9 system in which the original promoter replaced with a liver-specific promoter could still prevent HBV replication both in vitro and in vivo. Three functional guide RNAs (gRNAs), T2, T3, and T6, which target the conserved areas of different HBV genotypes, suggested consistently better anti-HBV results with different liver-specific promoters. In addition, the three gRNAs inhibited the replication of HBV genotypes A, B, and C to varying degrees. Under the action of the EnhII-Pa1AT promoter and AAV8, the expression of SaCas9 was additionally reduced in other organs or tissues compared to the liver. These data are helpful for clinical applications in the liver by ensuring that the results of the CRISPR/Cas9 system is kept refined to liver and, thereby, decreasing the possibility of disagreeable and injurious results through nonspecific targeting in other tissues [65].

## 8. Prevention of HCC via NAs—Which NA Is Better as First-Line Therapy in Terms of Preventing HCC?

A comparison involving ETV, TDF, and TAF on the reduction in HCC is summarized in Table 1.

The effectiveness of initial therapy with NAs, such as ETV, TDF, and TAF, in prohibiting HCC in CHB subjects remains unclarified. Several studies have produced opposite data concerning the influence of NAs on the risk and prohibition of HCC.

The debatable consequences can be, in part, owed to the arbitrary nature of significance levels, resulting in opposite evaluations from very similar datasets. The use of observational data, however, which is inclined to both the within- and between-study heterogeneity of subjects’ features, also brings additional uncertainty. The same-time adoption of ETV and TDF in East Asia, where most of these investigations were performed, additionally make more difficult analyses, as does the difference in the follow-up durations between the ETV and TDF groups. Clinicians performing meta-analyses in this section need to make many methodologic directions to make a mild bias but are fundamentally limited to the methodologies of the performed investigations. It is accordingly important for clinicians, as well as the readers of the issued meta-analyses, to be conscious of the quality of observational investigations and meta-analyses regarding the subject features, study direction, and statistical analyses [66,67,68,69,70,71,72,73,74,75,76,77,78,79,80,81] (Table 1).

It is critical to observe that all of the investigations comparing the risk of HCC between TDF and ETV treatment have given one direction supporting TDF or no direction. No high-quality investigations have brought evidence supporting ETV over TDF [82]. Additional clinical investigations or trials with a large number of subjects and a longer follow-up are essential to settle these opposite opinions and to obtain a consensus.

## 9. Conclusions

Thanks to the broad usefulness of the excellent efficacious antiviral therapy for CHB, long-term clinical results in subjects with CHB have been dramatically progressed over the previous ten years. Notwithstanding that recent antiviral therapy does not completely rule out the risk of HCC, any reducing occurrence of HBV-associated HCC remains to be evinced regarding the crude ratio and a definite number of subjects, unlike common expectations [12].

In addition, some patients at an immune-tolerant phase and at a high risk of HCC are still advised not to start antiviral therapy under recent therapy guidelines. A growing body of proof from big-scale group investigations demonstrates that early indications of antiviral therapy, even with continuously low ALT values, may be needed to minimize the risk of HCC.

In conclusion, these data demonstrate that elevated ALT levels should not be considered as important criteria to help the decision of starting antiviral therapy in patients with CHB [12].

The current cost-effective study has shown that earlier therapy starting with normal ALT values and a high viral value may be cost-efficacious compared to delaying the therapy until the occurrence of the active hepatitis stage in adult subjects with CHB [83], demonstrating the highly capable long-term effectiveness and safety of a recent anti-HBV study that posed a high genetic barrier to resistance and reducing the cost [84,85,86]. Accordingly, all drug adherence ratios in CHB subjects have been continuously demonstrated to be high (>90%) by many past investigations [87,88,89]. So, efforts to settle therapy guidelines with the current clinical results should be made to decrease the additional occurrence of HCC, such as the elongation of treatment policies and the careful choice of medicine, to optimize the HCC-prohibiting effect of the therapy [12].

Extending the treatment policy to CHB subjects with extreme values of HBV DNA may be thought to further prohibit HCC irrespective of the ALT values.

In patients with CHB treated with ETV or TDF, early ALT normalization has been independently related to a proportionally lower HCC risk, irrespective of steatosis or cirrhosis at the start and the VR during therapy [90].

Whether ETV, TDF, or TAF treatments differ in their effects on the prevention of HCC remains unclear. To resolve this issue, we suggest a meta-analysis with the use of individual subject data from group investigations or randomized studies with a bigger number of subjects and a longer follow-up.

## Figures and Tables

**Table 1 diagnostics-13-03212-t001:** Comparison of ETV, TDF, and TAF on reduction in HCC.

Study Area	Patients	Outcome	Superiority or Equality	Reference	Year
Korea national cohort Hospital cohort	ETV: 11,464 TDF: 12,692 ETV: 1560 TDF: 1141	ETV: 1.19/100 PY TDF: 0.89/100 PY HR 0.68, 95% CI 0.60–0.78 HR 0.68, 95% CI 0.46–0.99	TDF > ETV	Choi JG [66]	2019
Korea	ETV: 1484 TDF: 1413	ETV: 1.92/100 PY TDF: 1.69/100 PY HR 0.975, *p* = 0.852	ETV = TDF	Kim SU [67]	2019
Korea	ETV: 1583 TDF: 1439	HR 1.030, 95% CI 0.703–1.509 *p* = 0.880	TDF = ETV	Lee SW [68]	2020
Korea	ETV: 180 TDF: 224	HR 0.36, 95% CI 0.12–1.14 *p* = 0.08	ETV = TDF	Ha YJ [69]	2020
Korea	ETV: 813 TDF: 882	HR 0.82, 95% CI 0.68–0.98 *p* = 0.03 (after surgical resection)	TDF > ETV	Choi JG [70]	2021
Korea	ETV: 1525 TAF: 286	ETV: 1.67/100 PY TAF: 1.19/100 PY HR 0.681, 95% CI 0.351–1.320 *p* = 0.255	ETV = TAF	Lee HW [71]	2021
Korea	ETV: 1064 TDF: 629 TAF: 389	ETV: 1.45/100 PY TDF: 1.05/100 PY TAF: 0.65/100 PY *p* =0.340	ETV = TDF = TAF	Chon HY [72]	2021
Korea	TDF: 2245 TAF: 502	TDF: 0.90/100 PY TAF: 0.82/100 PY *p* = 0.60	TDF = TAF	Lim JH [73]	2022
China	ETV: 2124 TDF: 1574	RR 0.66, 95% CI 0.49–0.89, *p* = 0.008	TDF > ETV	Zhang Z [74]	2019
China	ETV: 28,041 TDF: 1309	ETV: 0.49/100 PY TDF: 0.06/100 PY HR 0.36, 95% CI 0.16–0.80, *p* = 0.013	TDF > ETV	Yip TC [75]	2020
Taiwan and Asia–Pacific	ETV: 4837 TDF: 700	HR 0.89; 95% CI 0.41–1.92, *p* = 0.77	TDF = ETV	Hsu YC [76]	2020
Taiwan and Asia–Pacific	ETV: 19,702 TDF: 16,266	ETV: 3.44%/5 Y TDF: 3.39%/5 Y HR 0.88, 95% CI 0.73–1.07 *p* = 0.20	TDF = ETV	Tseng CH [77]	2020
Hong Kong and China	ETV: 56,346 TDF: 28,662	HR 0.73, 95% CI 0.62–0.85 *p* < 0.001	TDF > ETV	Cheung KS [78]	2020
Europe	ETV: 772 TDF: 1163	ETV: 1.08/100 PY TDF: 1.2/100 PY *p* = 0.321	ETV = TDF	Papatheodoridis GV [79]	2020
USA	ETV: 2193 TDF: 1094	HR 1.00, 95% CI 0.76–1.32	ETV = TDF	Su F [80]	2021

ETV, entecavir; TDF, tenofovir disoproxil fumarate; TAF, tenofovir alafenamide fumarate; HCC, hepatocellular carcinoma; HR, hazard ratio; CI, confidence interval; PY, person years; RR, rate ratio—HR combined with incidence rate ratios; Y, year.

## Data Availability

Not applicable.

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
