# Peer review of "Hepatocellular Carcinoma and Hepatitis: Advanced Diagnosis and Management with a Focus on the Prevention of Hepatitis B-Related Hepatocellular Carcinoma"

_diagnostics, 2023, doi:10.3390/diagnostics13203212_

Round 1

Reviewer 1 Report

This is a well written review which examines the role of chronic HBV infection in the pathogenesis of HCC.

I have some concerns on a couple of points.

The paper is oriented mainly to Asian patients. This is a limitation for non-Asian readers and should be acknowledged  by the authors.  For example,  studies on the efficacy of ENT or TDF in preventing HCC are reported mainly from Korean groups and in only two studies from Europe and US (ref 77,78), both failing to show differences. A more accurate check of the literature on this point seems desirable and the results discussed  also in the light of patient origin.

In addition, there is no mention of the potential effects of the different HBV genotypes on liver disease progression and cancer development. There is a number of papers on this point, although not conclusive.

Good quality

Author Response

6 October 2023

Dear Sir,

Thank you for reviewing our manuscript titled “Hepatocellular Carcinoma and Hepatitis: Advanced Diagnosis and Management - Focus on prevention of hepatitis B virus-related hepatocellular carcinoma -” as ‘Review Article’ for publication submitted to your journal on August 25th.

Thank you for your valuable suggestions.

According to your suggestions, we have revised the manuscript and resubmit it.

We understand the papers regarding the comparison of NAs such as ETV, TAF and TDF on the prevention of HCC in Europe and the USA were not many so far. (reference No. 82)

So, we have described the above matter citing papers mainly performed in Asian countries including China, Korea and Japan.

Regarding the potential effects of the different HBV genotypes on liver disease progression and cancer development, and development of liver cancer, we have described the matter written by McMahon and Haga et al.

We attest that all the listed authors participated meaningfully in this study and have seen and approved the final manuscript.

We believe this manuscript is instructive and useful for general physicians especially hepatologists including pathologists engaged in the field of hepatic tumors.

We sincerely hope that this manuscript will be accepted by your journal.

Thank you very much for your kind support and consideration.

Sincerely yours,

Soo Ki Kim, M.D., Ph D.

Department of Gastroenterology, Kobe Asahi Hospital

3-5-25 Bououji-cho, Nagata-ku, Kobe, 653-0801, Japan

TEL +81-78-612-5151  FAX +81-78-612-5152

E-mail: kinggold@kobe-asahi-hp.com

Reviewer 2 Report

The article entitled "Hepatocellular Carcinoma and Hepatitis: Advanced Diagnosis and Management - Focus on prevention of hepatitis B virus-related hepatocellular carcinoma" submitted by Kim et al has been well written, however, have the following concern that needs to be focused on by the author:

1. It is important to focus on the genotype of HBV and its relevance to HCC management/treatment

2. There is a need to focus on the new technologies like CRISPR Cas and their potential for treating HBV-mediated HCC .

Author Response

6 October 2023

Dear Sir,

Thank you for reviewing our manuscript titled “Hepatocellular Carcinoma and Hepatitis: Advanced Diagnosis and Management - Focus on prevention of hepatitis B virus-related hepatocellular carcinoma -” as ‘Review Article’ for publication submitted to your journal on August 25th.

Thank you for your valuable suggestions.

According to your suggestions, we have revised the manuscript and resubmit it.

Regarding the genotype of HBV and its relevance to HCC management/treatment, we have described the matters citing by McMahon BJ and Haga H’s papers.

Regarding the new technologies like CRISPR/Cas9 and potential for treating HBV-mediated HCC, we have introduced a paper written by Yan K, et al.

We attest that all the listed authors participated meaningfully in this study and have seen and approved the final manuscript.

We believe this manuscript is instructive and useful for general physicians especially hepatologists including pathologists engaged in the field of hepatic tumors.

We sincerely hope that this manuscript will be accepted by your journal.

Thank you very much for your kind support and consideration.

Sincerely yours,

Soo Ki Kim, M.D., Ph D.

Department of Gastroenterology, Kobe Asahi Hospital

3-5-25 Bououji-cho, Nagata-ku, Kobe, 653-0801, Japan

TEL +81-78-612-5151  FAX +81-78-612-5152

E-mail: kinggold@kobe-asahi-hp.com

Round 2

Reviewer 2 Report

the manuscript can be accepted